# Childhood Predictors of Healthcare Use and Health Status in Early Adulthood: Findings from the Twins Early Development Study

**DOI:** 10.3390/ijerph192316349

**Published:** 2022-12-06

**Authors:** Paul McCrone, Janet Boadu

**Affiliations:** 1Institute of Lifecourse Development, University of Greenwich, London SE10 9LS, UK; 2King’s Health Economics, Institute of Psychiatry, Psychology and Neuroscience, King’s College London, London WC2R 2LS, UK

**Keywords:** service use, cognition, twins, demand for health

## Abstract

The use of healthcare services is likely to be associated with need but also the factors relating to the care system and the ability to negotiate around it. Healthcare use and health status may also be associated with the factors that exist in childhood. This study aims to identify the demographic, clinical, and cognitive characteristics of children at age 4 that impact healthcare use and health status at age 21. The data from the Twins Early Development Study were used. Health problems, healthcare use, and cognitive ability at age 4 were entered into generalised estimating equations to predict the use of general practitioners, outpatient services, counselling, emergency clinic visits, and a healthcare helpline at age 21. Similar models existed for the prediction of whether problems were recorded on the EQ-5D-5L EuroQol instrument. The data on up to 6707 individuals were available for analysis. Sex was a significant predictor of service use, with boys being more likely than girls to later use all services, except for emergency clinic visits. Certain health conditions at age 4 predicted the use of services with models differing according to service type. Greater general cognitive ability predicted higher use of general practitioners, outpatient care, and health helplines. The current health status was strongly predictive of service use. Service use in young adulthood was significantly related to concurrent health status as well as health conditions in childhood. General cognitive ability was significantly associated with the use of general practitioner contacts, outpatient visits, and the use of a health helpline.

## 1. Introduction

The use of healthcare services will usually be a response to the existence of health problems. How strong the link is will vary, and the use of services will likely also depend on other factors which are not necessarily health indicators in themselves. Even after taking health status into account, there will be differences in use representing inequalities in access to care. Influential work by Andersen and Newman established a theoretical model for the use of healthcare services and identified three key types of factors that could influence this: predisposing features (e.g., age, sex, ethnicity, previous illnesses, etc.), enabling features (e.g., income, access to care, employment, etc.), and needs (e.g., symptoms, diagnosis, functioning, etc.) [1]. Adaptations to the model have been widespread [2], and a review of longitudinal studies using the model is underway [3]. A further review has found that, while the model has mainly been applied in quantitative studies, it also has relevance for qualitative research [4].

It may be expected, and indeed desired, that greater healthcare use occurs in those with the greatest needs. However, as Tudor Hart suggested 50 years ago, an inverse relationship exists between use and need [5]. Health inequalities are key features of nations, and these are not fully addressed just by having well-funded, free at-the-point-of-delivery services [6]. Other work looking at the demand for health has suggested that education may be positively associated with the receipt of care [7]. This may be reflecting greater returns on education in the form of income, which enables easier access. However, in a free at-the-point-of-use service, this is not as important (although, as mentioned above, inequalities still exist when services are not paid for when used). What has not been investigated to any great extent is whether education is reflecting cognitive ability or intelligence which may facilitate easier access to care in what to some may be a complex system.

One area of work that is relatively undeveloped is that of the childhood predictors of long-term healthcare use and health status. Some health problems will emerge in the early years of childhood and may persist into adulthood. Other problems may be transient, but whether or not their occurrence can specifically explain subsequent service use is unclear. Their occurrence may reflect other aspects of life that themselves are predictors of future service use. The literature on these relationships is limited and tends to focus on the follow-up of children with specific conditions such as depression or antisocial behaviour [8,9]. Studies that identify the predictors from general population samples are lacking.

Cohort studies provide an ideal opportunity to identify the predictors of long-term service use and health status. In 1994, the Twins Early Development Study (TEDS) began and has, over time, followed up a large cohort of twins from across the UK [10]. Using the data from that study, we aim to identify the demographic and health-related factors in childhood that predict (i) healthcare use and (ii) health status in early adulthood.

## 2. Materials and Methods

### 2.1. Sample

Data were obtained from TEDS [10]. This included twins who were born between 1994 and 1996 in England and Wales and had as its focus the cognitive and behavioural development of the participants over time. TEDS originally included 16,810 twin pairs. Informed consent was obtained from the parents of all the participants, and the study was approved by the Institute of Psychiatry (King’s College London) Ethics Committee. After informed consent was provided, data were collected at regular intervals on a range of cognitive and behavioural characteristics and health issues in childhood. At age 21, data were collected on health and social service use and on health status. The information collected at age 4 was provided by parents or via parent-administered tests.

### 2.2. Measures

The data collected when the participants were aged around four years old were used in the analyses. The variables included demographic details (age, gender, ethnicity, parental marital status), a measure of socioeconomic status (a composite measure derived from parental qualifications, mother’s age on the birth of their first child, and parental occupation, with higher scores representing higher status), health problems when four years old (bacterial meningitis, skin conditions, hearing problems, talking problems, asthma, eyesight problems, stomach ache, vomiting, weakness/paralysis, having fits), developmental issues (autism, learning disability), use of health services at four years old (general practitioner contacts, visits to accident and emergency department, admission to hospital, surgical operation), and the subscales of the Strengths and Difficulties Questionnaire (emotional symptoms, conduct problems, hyperactivity/inattention, peer relationship problems, prosocial behaviour, with higher scores representing greater severity) [11]. It will be seen that the potential predictors related to childhood health issues are varied and includes diseases such as asthma (which usually will continue into adulthood), illnesses that may be short-term (for example, bacterial meningitis), and symptoms (for example, stomach aches). The analyses were essentially exploratory, and we did not derive a priori hypotheses regarding these.

The general cognitive ability was a composite variable based on two measures of verbal performance and two of non-verbal performance [12]. The composite measure was standardised with a mean of zero and a variance of one.

Service use during the 12 months prior to the most recent round of data collection at age 21 was recorded with an adapted version of the Client Service Receipt Inventory (CSRI) [13]. The respondents were asked whether they had used specific services, and these included general practitioner contacts, visits to accident and emergency departments, counsellor contacts, calls to the NHS helpline, and outpatient visits.

The five-level version of the EuroQol instrument (EQ-5D-5L) was used to assess the health-related quality of life [14]. This measure covers five domains: mobility, self-care, usual activities, pain/discomfort, and anxiety/depression. Each domain receives an integer score of 1 (no problem in that area) to 5 (extreme problems).

### 2.3. Analysis

A series of generalised estimating equations (GEEs) were used to assess the impact of background characteristics on the dependent variables representing service use and the health-related quality of life indicators following the recommendations from Carlin et al. [15]. The dependent variables were binary, scoring 1 if a service was used and 0 otherwise. The EQ-5D-5L was also recoded into binary variables, with 1 representing a problem in the particular domain and 0 representing no problem. The account was taken of the potential correlation between individuals in each twin pair using the xtgee command in Stata and with robust stand errors reported. The models assumed a binomial family distribution.

In addition to the background variables listed earlier, we also included the binary variables derived from the EQ-5D-5L in the analysis of service use to account for current morbidity. In the analysis of each of the five health status domains, we included the four binary variables indicating problems or not in the four other domains. All the variables were entered into the models in a single block, odds ratios were reported, and statistical significance was indicated with 95% confidence intervals.

## 3. Results

### 3.1. Sample Characteristics

The total dataset consisted of 22,248 individuals. There were data for children from age 4, and the service use data at age 21 were available for 6707 (30.1%) of these. For some analyses, there were fewer included participants due to missing data, and this is indicated in subsequent tables. Just over one-third of the participants were male, and the vast majority were of White ethnicity (Table 1). At age 4, nearly three-quarters had seen a GP, and 15% had visited A&E. The most common health problems or conditions at that age were skin conditions, talking problems, hearing problems, and asthma. For most participants, their parents were married or cohabiting. Scores on continuous measures are shown in Table 2.

At age 21, the EQ-5D-5L revealed that anxiety/depression problems were experienced by 40% of the participants, and one-quarter had problems with pain/discomfort (Table 3). Self-care problems were the least likely to occur. The most frequently used service was GP contacts, with over two-thirds of the respondents reporting this (Table 4). About one-quarter of participants had outpatient appointments at age 21, and one-fifth had attended A&E. Contact with NHS Direct and the use of counsellors both occurred in fewer than 15% of the participants.

### 3.2. Correlations and Predictors of Service Use

Compared with females, males were less likely to have had contact with GPs during the 12 months prior to long-term follow-up (Table 3). Those who had seen a GP in the period prior to the 4-year assessment were more likely to have had GP contacts at follow-up. A greater socioeconomic score was associated with increased GP use, as was greater cognitive ability at 4 years old. Having current problems with usual activities, pain/discomfort, and anxiety/depression were all associated with increased GP use, as was having asthma when 4 years old. Those participants who had a single parent as 4-year-olds were subsequently more likely to have GP contacts than those participants who had parents who were married or cohabiting.

Accident and emergency contacts at long-term follow-up were more likely for those participants with greater SDQ conduct scores at 4 years old, greater SDQ prosocial scores at 4 years old, current problems with usual activities, current problems with pain/discomfort, and asthma as 4-year-olds, as well as those having parents who were single.

Contact with counsellors at long-term follow-up was more likely for females, those who had been diagnosed with autism by 4 years of age, those with greater SDQ anxiety scores, higher socioeconomic scores, and current problems with self-care, usual activities, pain/discomfort, and anxiety/depression.

Those who contacted NHS Direct at long-term follow-up were more likely to be female, to not have had a skin condition at 4 years old, to have had higher SDQ prosocial scores at 4 years old, to have greater cognitive abilities at 4 years old, current mobility problems, problems with usual activities, problems with pain/discomfort, problems with anxiety/depression, and having asthma at 4 years old.

Finally, long-term contact with outpatient services was again significantly more likely for females. This was also more likely if, as a 4-year-old, the participant had had bacterial meningitis; if they had been admitted to the hospital at that age; if they had higher SDQ prosocial scores, higher socioeconomic scores, or greater cognitive ability at 4 years old; if they had current problems with mobility, usual activities, or pain/discomfort; and if they had suffered from stomach aches at 4 years old.

### 3.3. Correlates and Predictors of Health Status

The models that identify the significant predictors or correlations of problems at age 21 in the domains of the EQ-5D-5L are shown in Table 5. Problems with mobility were more likely for males than females. They were also more likely if the child had attended A&E at age 4 but less likely if they had been admitted to the hospital. Lower socioeconomic status was associated with less likelihood of mobility problems. The problems in this domain were more likely if there were also problems with self-care, usual activities, or pain/discomfort.

Current problems with self-care were more likely if the child had a skin condition at age 4. They were also more likely if the respondent had current problems with mobility, usual activities, or anxiety/depression.

Problems at age 21 with usual activities were less likely if the child had attended A&E at age 4, or if they had experienced fits. They were more likely if they had eyesight problems at age 4, and if they had current problems with mobility, self-care, pain/discomfort, or anxiety/depression.

Current problems with pain/discomfort were more likely if the child had asthma at age 4. They were also more likely if, at age 21, they also had problems with mobility, usual activities, or anxiety/depression.

Finally, anxiety/depression problems at age 21 were more likely for females and for those who had autism at age 4 but less likely for those who had asthma. Higher scores on the SDQ anxiety and conduct subscales predicted current anxiety/depression problems. Problems with usual activities or pain/discomfort were associated with anxiety/depression problems. Lastly, if the child had a single parent, then they were more likely to have anxiety/depression problems than if their parent was married or cohabiting.

## 4. Discussion

This study showed that a range of demographic characteristics, childhood health, and current functioning are significant predictors of the use of healthcare services when aged 21. Current health status, as measured with the EQ-5D-5L, was strongly associated with the use of healthcare services. This was particularly the case for usual activities and pain/discomfort which predicted higher use of all the five services included in the analyses. The most consistent childhood demographic factor was sex, with boys having less use of GPs, counsellors, NHS Direct, and outpatient care than girls. This is consistent with other studies that suggest that men use healthcare services less than women. Other demographic characteristics were significant predictors of the use of some services and not others. There did appear to be a persistence of GP use, with the use at age 4 predicting the use in early adulthood.

We found that higher levels of cognition at age four were predictive of the greater use of GPs, outpatient care, and NHS Direct in adulthood. Given that this was after controlling for current health status and childhood health problems, cognition has a lasting impact on access to care, and this may be through its impact on health literacy. Other studies have reported the opposite effects of intelligence. In an analysis of a sample of representative people from Luxembourg, Wrulich et al. examined the impact of childhood intelligence on health status in adulthood [16]. They found that higher intelligence in adolescence was associated with fewer doctor visits and better functional health when the median age was 51.7 years.

In our analyses, it is interesting that higher cognitive ability was not a significant predictor of lower use of any of the services. Following the model of Andersen and Newman [1], we would expect use to be related to need, and the analyses do demonstrate this. The additional impact of cognitive ability indicates that this may be required to negotiate what for some may be a complex system in order to access care.

Certain conditions that were present at the age of four were predictive of future service use. Asthma was associated with the use of GPs, visits to accident and emergency departments, and calls to NHS Direct in early adulthood. This was unsurprising, given that asthma often emerges in childhood and is, for many, a chronic condition. Having autism recognised at age four was associated with the future use of counsellors, and this also might be unsurprising. It is harder to understand why having bacterial meningitis or stomach aches were associated with future outpatient use, or why skin conditions were associated with less use of NHS Direct. These could of course be spurious findings given the large number of tests conducted (see limitations below).

There were limitations to this study. First, we did not have a comparison group, and this is a common issue with cohort studies. The inclusion of a comparison was not feasible but would potentially have strengthened the findings. Second, in our multivariable analyses, we used a fairly simple model without interaction terms. There may well be interaction effects that are relevant to explore, such as the link between SES and cognition. Third, our analyses were somewhat exploratory in that, for some variables, there was no a priori expectation of a relationship with service use or quality of life. By chance, we would expect some of these relationships to be statistically significant (and no adjustment was made to significance levels).

## 5. Conclusions

Health service use and health status in early adulthood were shown to be related to a number of childhood characteristics. Cognitive ability was associated with the use of general practitioner contacts, outpatient visits, and the use of health helplines, and this may indicate challenges in navigating through the healthcare system.

## Figures and Tables

**Table 1 ijerph-19-16349-t001:** Sample characteristics (categorical variables).

Variable	N	%
Male sex	2454	36.6
White ethnicity	6330	94.6
Parents’ marital status		
Married/cohabiting	6316	95.2
Divorced or separated	63	1.0
Widowed	118	1.8
Single	138	2.1
Service use by age 4		
Seen GP	4712	70.4
Admitted to hospital	306	4.6
A&E	1040	15.5
Surgical operation	271	4.1
Health problems/conditions by age 4		
Bacterial meningitis	61	0.9
Autism	95	1.4
Skin condition	2664	39.8
Hearing problems	1301	19.4
Talking problems	1332	19.9
Asthma	1143	17.1
Eyesight problems	420	6.3
Stomach ache	363	5.4
Vomiting	155	2.3
Weakness/paralysis	49	0.73
Fits	66	1.0
Learning disability	124	1.9

**Table 2 ijerph-19-16349-t002:** Sample characteristics (continuous variables).

Variable	Mean	SD
Age	4.02	0.13
Socioeconomic status	0.32	0.98
Strengths and Difficulties Questionnaire subscores		
Anxiety	1.37	1.42
Conduct	1.92	1.51
Hyperactivity	3.74	2.27
Prosocial	7.42	1.86
Peer problems	1.43	1.46
Cognition	0.09	0.94

**Table 3 ijerph-19-16349-t003:** Number (%) of respondents reporting service use in previous year and problems in EQ-5D-5L domains.

Variable	N	%
Service use (y/n)		
Outpatient	1569	23.4
GP	4663	69.7
A&E	1392	20.8
Counsellor	822	12.3
NHS Direct	967	14.5
Problem on EQ-5D-5L domain (y/n)		
Mobility	485	7.3
Self-care	157	2.4
Usual activities	778	11.7
Pain/discomfort	1686	25.4
Anxiety/depression	2655	40.0

N = sample number, GP = general practitioner, A&E = accident and emergency department, NHS = National Health Service.

**Table 4 ijerph-19-16349-t004:** Generalised estimating equations showing predictors of service use at age 21.

	GP			A&E			Counsellor			NHS Direct			Outpatient		
		95% CI			95% CI			95% CI			95% CI			95% CI	
Variable (at Age 4 Unless Stated Otherwise)	OR	LL	UL	OR	LL	UL	OR	LL	UL	OR	LL	UL	OR	LL	UL
Demographic															
Male sex	**0.39**	**0.34**	**0.44**	1.02	0.88	1.18	**0.53**	**0.43**	**0.65**	**0.50**	**0.42**	**0.61**	**0.74**	**0.64**	**0.86**
Age	0.91	0.51	1.63	1.13	0.67	1.92	1.60	0.90	2.84	1.49	0.82	2.68	1.30	0.82	2.04
White ethnicity	1.19	0.90	1.58	1.18	0.85	1.65	0.88	0.58	1.33	1.39	0.92	2.10	1.17	0.85	1.60
Socioeconomic	**1.10**	**1.03**	**1.18**	0.95	0.89	1.03	**1.29**	**1.17**	**1.41**	1.00	0.91	1.09	**1.07**	**1.00**	**1.15**
Service use															
Seen general practitioner	**1.17**	**1.02**	**1.34**	1.10	0.94	1.28	1.05	0.86	1.28	1.01	0.84	1.20	1.09	0.94	1.26
Admitted to hospital	1.24	0.90	1.71	1.23	0.89	1.71	1.00	0.62	1.59	1.24	0.84	1.83	**1.69**	**1.23**	**2.31**
Emergency department	0.96	0.81	1.14	1.11	0.93	1.33	1.03	0.80	1.33	0.95	0.76	1.19	0.95	0.79	1.14
Surgical operation	0.90	0.65	1.24	0.92	0.64	1.31	1.11	0.69	1.78	0.93	0.60	1.44	1.04	0.74	1.48
Health problems															
Asthma	**1.45**	**1.23**	**1.72**	**1.20**	**1.01**	**1.44**	0.86	0.67	1.11	**1.34**	**1.10**	**1.63**	1.16	0.98	1.37
Autism	1.61	0.86	3.02	1.16	0.65	2.09	**2.05**	**1.07**	**3.95**	1.27	0.63	2.58	0.67	0.35	1.28
Bacterial meningitis	1.36	0.59	3.11	1.42	0.75	2.67	1.27	0.48	3.40	1.83	0.86	3.88	**2.19**	**1.18**	**4.06**
Eyesight problems	1.10	0.84	1.42	0.98	0.74	1.30	1.22	0.86	1.72	1.20	0.87	1.64	1.11	0.85	1.44
Fits	0.63	0.32	1.22	1.29	0.62	2.69	0.56	0.15	2.06	1.18	0.55	2.55	1.17	0.62	2.22
Hearing problems	1.14	0.97	1.34	1.05	0.87	1.25	0.93	0.73	1.18	0.99	0.80	1.22	1.10	0.92	1.30
Learning disability	0.81	0.51	1.28	0.96	0.55	1.67	0.59	0.22	1.55	0.56	0.24	1.29	1.27	0.73	2.21
Skin condition	1.04	0.91	1.18	1.01	0.88	1.15	1.08	0.90	1.29	**0.81**	**0.69**	**0.96**	0.92	0.80	1.05
Stomach ache	1.07	0.79	1.46	1.08	0.79	1.46	1.02	0.70	1.49	1.22	0.86	1.74	**1.36**	**1.02**	**1.81**
Talking problems	0.99	0.84	1.16	0.94	0.78	1.14	1.22	0.96	1.56	1.06	0.85	1.32	0.94	0.79	1.13
Vomiting	1.05	0.68	1.60	1.46	0.96	2.22	1.49	0.85	2.62	1.33	0.82	2.16	1.17	0.74	1.84
Weakness/paralysis	0.93	0.42	2.05	0.78	0.33	1.87	0.47	0.09	2.45	0.72	0.26	1.96	0.92	0.41	2.06
SDQ subscores															
Anxiety	1.02	0.97	1.07	0.98	0.93	1.03	**1.09**	**1.02**	**1.16**	0.99	0.93	1.05	0.99	0.94	1.04
Conduct	1.02	0.98	1.08	**1.06**	**1.01**	**1.12**	0.99	0.92	1.06	1.01	0.95	1.07	1.04	0.99	1.09
Hyperactivity	1.01	0.98	1.04	0.99	0.96	1.02	0.98	0.94	1.02	1.02	0.98	1.06	1.03	1.00	1.07
Prosocial	1.03	1.00	1.07	**1.06**	**1.02**	**1.11**	0.97	0.92	1.03	**1.05**	**1.00**	**1.10**	**1.05**	**1.01**	**1.09**
Peer problems	1.00	0.95	1.04	0.99	0.94	1.04	1.06	0.99	1.13	1.05	0.99	1.11	1.01	0.96	1.06
General cognitive ability	**1.11**	**1.03**	**1.20**	0.99	0.91	1.07	1.03	0.93	1.14	**1.13**	**1.03**	**1.25**	**1.11**	**1.03**	**1.20**
Parents’ marital status															
Married/cohabiting	Reference category			Reference category			Reference category			Reference category			Reference category		
Divorced or separated	1.65	0.71	3.84	1.48	0.58	3.76	0.46	0.12	1.81	1.13	0.33	3.84	1.52	0.62	3.74
Widowed	0.90	0.54	1.50	0.93	0.54	1.60	0.90	0.44	1.83	1.38	0.81	2.35	1.34	0.81	2.23
Single	**2.28**	**1.13**	**4.62**	**2.31**	**1.33**	**4.00**	1.31	0.56	3.08	1.23	0.59	2.58	1.51	0.83	2.73
Health problems at age 21															
Mobility problems	1.07	0.79	1.45	1.29	0.99	1.69	0.91	0.64	1.29	**1.41**	**1.05**	**1.89**	**1.85**	**1.45**	**2.37**
Self-care problems	1.41	0.76	2.64	1.20	0.79	1.83	**1.88**	**1.20**	**2.93**	0.91	0.57	1.45	1.36	0.92	2.01
Usual activities problems	**1.83**	**1.41**	**2.36**	**1.29**	**1.03**	**1.61**	**2.64**	**2.06**	**3.38**	**1.60**	**1.25**	**2.04**	**1.32**	**1.07**	**1.62**
Pain/discomfort problems	**1.56**	**1.34**	**1.83**	**1.54**	**1.32**	**1.81**	**1.26**	**1.02**	**1.55**	**1.43**	**1.19**	**1.72**	**1.86**	**1.61**	**2.16**
Anxiety/depression problems	**1.49**	**1.31**	**1.69**	1.15	1.00	1.32	**4.68**	**3.85**	**5.68**	**1.25**	**1.06**	**1.47**	1.08	0.94	1.23
Constant term	1.68	0.16	18.00	0.05	0.01	0.49	0.01	<0.01	0.10	0.01	<0.01	0.15	0.04	0.01	0.25
Sample size	5627			5633			5624			5624			5629		

Note: Figures in bold represent statistically significant findings at *p* < 0.05 level. N = sample number, OR = odds ratio, LL = lower limit of confidence interval, UL = upper limit of confidence interval, GP = general practitioner, A&E = accident and emergency department, NHS = National Health Service.

**Table 5 ijerph-19-16349-t005:** Generalised estimating equations showing predictors of problems on EQ-5D-5L domains at age 21.

	Mobility			Self-Care			Usual Activities			Pain/Discomfort			Anxiety/Depression		
		95% CI			95% CI			95% CI			95% CI			95% CI	
Variable (at Age 4 Unless Stated Otherwise)	OR	LL	UL	OR	LL	UL	OR	LL	UL	OR	LL	UL	OR	LL	UL
Demographic															
Male gender	**1.32**	**1.00**	**1.73**	0.74	0.45	1.22	0.89	0.71	1.11	0.92	0.79	1.07	**0.61**	**0.54**	**0.69**
Age	0.95	0.28	3.25	0.78	0.07	8.73	0.68	0.30	1.51	0.95	0.59	1.55	1.33	0.84	2.11
White ethnicity	0.98	0.55	1.76	1.38	0.50	3.82	0.86	0.55	1.33	1.02	0.73	1.42	1.04	0.79	1.37
Socioeconomic	**0.86**	**0.75**	**0.99**	1.09	0.89	1.33	1.09	0.98	1.21	1.03	0.96	1.11	1.03	0.97	1.09
Service use															
Seen general practitioner	1.17	0.88	1.56	1.08	0.67	1.75	0.82	0.66	1.02	1.00	0.86	1.17	0.99	0.87	1.12
Admitted to hospital	**0.50**	**0.25**	**0.99**	1.13	0.41	3.07	1.39	0.82	2.35	1.34	0.97	1.86	0.77	0.56	1.04
Emergency department	**1.73**	**1.24**	**2.41**	0.80	0.41	1.55	**0.69**	**0.51**	**0.94**	0.86	0.71	1.04	1.06	0.91	1.24
Surgical operation	1.00	0.47	2.12	1.59	0.61	4.15	0.91	0.52	1.57	0.83	0.58	1.21	1.33	0.97	1.83
Health problems															
Asthma	0.88	0.63	1.23	1.26	0.75	2.11	1.29	1.00	1.67	**1.22**	**1.02**	**1.45**	**0.82**	**0.70**	**0.96**
Autism	1.11	0.34	3.59	1.77	0.57	5.47	1.46	0.66	3.25	0.82	0.45	1.51	**2.93**	**1.71**	**5.03**
Bacterial meningitis	1.32	0.30	5.85	1.24	0.10	15.33	1.30	0.35	4.75	0.55	0.23	1.28	1.16	0.61	2.19
Eyesight problems	1.29	0.79	2.11	0.65	0.26	1.61	**1.58**	**1.08**	**2.29**	0.83	0.61	1.12	1.09	0.86	1.38
Fits	1.23	0.44	3.48	3.80	0.21	70.10	**0.10**	**0.01**	**0.79**	1.12	0.55	2.25	0.86	0.48	1.54
Hearing problems	0.97	0.70	1.36	1.21	0.69	2.11	0.98	0.74	1.29	1.02	0.86	1.22	1.01	0.87	1.18
Learning disability	0.62	0.20	1.93	2.05	0.43	9.81	0.60	0.25	1.42	0.98	0.56	1.73	0.85	0.52	1.37
Skin condition	0.82	0.63	1.06	**1.65**	**1.08**	**2.51**	1.00	0.81	1.23	1.10	0.96	1.26	1.11	0.98	1.24
Stomach ache	0.59	0.31	1.10	0.32	0.06	1.80	0.94	0.60	1.46	1.15	0.85	1.56	1.13	0.87	1.47
Talking problems	1.04	0.75	1.46	1.12	0.63	1.97	0.95	0.72	1.25	0.99	0.82	1.19	1.00	0.85	1.17
Vomiting	0.85	0.33	2.20	1.00	-	-	1.17	0.60	2.28	1.11	0.69	1.79	0.83	0.57	1.22
Weakness/paralysis	2.78	0.75	10.30	1.45	0.40	5.24	1.43	0.50	4.14	0.57	0.20	1.57	0.90	0.46	1.77
SDQ subscores															
Anxiety	0.99	0.89	1.09	1.10	0.95	1.27	0.98	0.91	1.06	1.03	0.98	1.08	**1.09**	**1.04**	**1.14**
Conduct	0.99	0.91	1.09	1.16	0.98	1.38	1.02	0.95	1.10	1.01	0.96	1.06	**1.07**	**1.02**	**1.12**
Hyperactivity	0.97	0.91	1.04	0.96	0.86	1.06	1.00	0.95	1.06	1.01	0.97	1.04	0.99	0.96	1.02
Prosocial	0.98	0.91	1.05	1.12	0.98	1.28	1.01	0.95	1.06	0.99	0.95	1.03	1.00	0.97	1.04
Peer problems	1.00	0.91	1.10	1.03	0.88	1.20	1.02	0.95	1.10	0.99	0.94	1.04	1.03	0.98	1.07
General cognitive ability	1.03	0.89	1.20	0.86	0.68	1.10	0.95	0.84	1.07	1.01	0.93	1.09	0.98	0.92	1.05
Parents’ marital status															
Married/cohabiting	Reference category			Reference category			Reference category			Reference category			Reference category		
Divorced or separated	0.57	0.18	1.81	1.00	-	-	0.78	0.28	2.13	1.78	0.92	3.46	1.02	0.51	2.06
Widowed	1.71	0.69	4.23	0.51	0.09	3.01	**2.17**	**1.08**	**4.34**	0.85	0.45	1.58	0.92	0.56	1.53
Single	1.33	0.32	5.64	2.25	0.36	14.21	0.48	0.13	1.82	0.64	0.29	1.41	**2.33**	**1.28**	**4.25**
Health problems at age 21															
Mobility problems	-	-	-	**3.26**	**2.00**	**5.32**	**7.41**	**5.59**	**9.81**	**10.92**	**8.20**	**14.55**	0.81	0.62	1.06
Self-care problems	**3.21**	**1.93**	**5.33**	-	-	-	**23.80**	**13.08**	**43.32**	1.25	0.73	2.15	1.66	1.00	2.75
Usual activity problems	**7.15**	**5.45**	**9.37**	**24.09**	**13.21**	**43.96**	**-**	**-**	**-**	**2.71**	**2.19**	**3.36**	**3.59**	**2.93**	**4.39**
Pain/discomfort problems	**11.10**	**8.33**	**14.80**	1.51	0.89	2.56	**2.80**	**2.26**	**3.46**	**-**	**-**	**-**	**1.56**	**1.36**	**1.79**
Anxiety/depression problems	1.03	0.80	1.33	**2.23**	**1.34**	**3.71**	**3.83**	**3.12**	**4.69**	**1.57**	**1.36**	**1.80**	-	-	-
Constant term	0.02	<0.01	2.74	<0.01	<0.01	20.00	0.16	0.01	4.33	0.23	0.03	1.68	**0.14**	**0.02**	**0.92**
Sample size	5641			5483			5641			5641			5641		

Note: Figures in bold represent statistically significant findings at *p* < 0.05 level.

## Data Availability

TEDS data are available at reasonable request from https://teds.ac.uk/researchers/teds-data-access-policy (accessed on 5 December 2022).

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
