# Peer review of "Childhood Predictors of Healthcare Use and Health Status in Early Adulthood: Findings from the Twins Early Development Study"

_ijerph, 2022, doi:10.3390/ijerph192316349_

Round 1

Reviewer 1 Report

Needs some work on the design. Cohort Studies without controls are of lower evidence strength than those with controls.

- Sample seems biased to Low SES (see mean score). SES is a significant issue when access and other issues that determine cognition are involved. Cognition can be ecologically dependent. Thus "intelligence" is an inappropriate variable without adequate caveats. A separation of Lower-SES from Higher SES cohorts may provide comparisons for more precise analyses and conclusions

Caveats are needed with regard to sampling (several significantly low proportions of certain groups may need consideration when aggregated. Aggregated effects may become significant.

Needs a conclusion section for clarity

Table 4 needs reconfiguration for clarity

Results need more discussion for clarity - e.g.: table 4 indicates, less service use for male sex but overall more GP use at age 4.

Proof read for some grammar, spelling and other errors.

Feedback comments are on the reviewed document

Author Response

Thank you for your really helpful comments. We have responded as follows:

  1. Needs some work on the design. Cohort Studies without controls are of lower evidence strength than those with controls We agree with the referee on this point. We do not have a comparison group and so have included this as a limitation (page 7).
  2. Sample seems biased to Low SES (see mean score). SES is a significant issue when access and other issues that determine cognition are involved. Cognition can be ecologically dependent. Thus "intelligence" is an inappropriate variable without adequate caveats. A separation of Lower-SES from Higher SES cohorts may provide comparisons for more precise analyses and conclusions. We do include SES as a potential predictor of service use and quality of life, but not in interaction with other variables. This is a limitation that we refer to on page 7.
  3. Caveats are needed with regard to sampling (several significantly low proportions of certain groups may need consideration when aggregated. Aggregated effects may become significant. The proportions certainly are low for some categories but the numbers are still high even in these categories. We did exclude some categories where the numbers were very low.
  4. Needs a conclusion section for clarity This has now been included (page 7).
  5. Table 4 needs reconfiguration for clarity Table 4 and other tables have bene formatted by the journal and do appear quite different to the Word document we uploaded. [Note to editors - can you look into this and make the tables more readable?] We have include a key for the abbreviations used
  6. Results need more discussion for clarity - e.g.: table 4 indicates, less service use for male sex but overall more GP use at age 4 The relationship between sex and GP use was mentioned in the Results and Discussion. We have also included something on the relationship between GP use at age 4 and in early adulthood in the Discussion.
  7. Proof read for some grammar, spelling and other errors We have done and this and corrected errors we found.
  8. Feedback comments are on the reviewed document We could not see any further comments - are there some?

Author Response

Thank you for your really helpful comments. We have responded as follows:

  1. Introductory remarks We thank the referee for these points and for recognising the importance of this work.
  2. Other papers investigating the model We thank the referee for drawing our attention to the other reviews. We have now referred to these (page 3).
  3. Tudor-Hart comment Yes, this is now an old reference but still of course highly influential. The book by Bartley is a great suggestion and we have cited this now (page 3).
  4. I didn't catch the idea of not ending the sentence starting with the words " Subsequent work has supported......" We agree that this made no sense! We have corrected the text.
  5. Issue about chronic and transient disorders The comment was not all that clear but we think we understand and have added some text relating to this on page 3.
  6. More explanations are needed, I think about the composition of variables This is a good point. To be honest, we included some variables where we would expect long-term relationships and others where this was very much exploratory. We have been clear now about this on page 4 and in the Discussion we refer to this issue. 
  7. The part titled 'Analysis" should also be reshaped. It is clear only to people with full knowledge of the Andersens model. But the majority of people even did not know o about Mr Andersen. The analysis section doe snot refer to the model but it should apparent from the preceding revised text that this is what is being referred to. 
  8. Here are a few features of a well-prepared paper Thank you for this point. We hope we have adhered to the correct structure of the paper.
  9. Don't You think that is necessary to use the room for more exciting information instead that 1,8% of respondents were widowed: or 1,8% were divorced or separated? There are names "other" for such situations. If this was in the text we would agree, but containing this information in tables does not seem problematic. We agree that there are lots of tests and we have referred to this in the Discussion.
  10. I believe that is not necessary to prepare such a table as table no.4. It is possible to say about issues over there "in words" and have the original "on request". Table 4 is key to the whole paper and so we prefer to keep it. However, we are happy to discuss with the Editor whether it should be included as supplementary information.
  11. Points 3.2 and 3.3. have the titles word "Predictors " I would be very grateful if the authors thought it over, including the same in the paper title. Correlations instead the predictors. And "for predictors" is the place in eventually in the discussion. Technically most of the independent variables can be considered to be predictors because they were measured many years prior to health use and health status. However, we do include health status in the health use model and vice versa and these do represent correlations. We have revised the text accordingly.
  12. The Paper hasn't had conclusions /recommendations We have now included a Conclusions section

Reviewer 3 Report

This is a well-written, clear paper that makes use of a strong data source to help healthcare decision makers predict - and potentially mitigate against - future resource use. The methods are broadly sound, being based on well-known regression techniques.

My only suggestion to improve the paper is to be careful about how some of the findings are reported. Using 95% significance means that there is still a potential 5% likelihood that the findings can be explained by chance. Therefore, with so many interactions and variables included, it is likely that SOME of the reported 'significant' findings might be a result of chance rather than genuine correlation. Instead, it would be better for the authors to report p values (to give more information about the confidence) and also avoid using phrases such as "...WERE significantly less likely to...". It would also be helpful to add a sentence to the discussion to highlight the potential role of chance in some of the findings.

Otherwise, the paper is excellent.

Author Response

Thank you for your really helpful comments. We have responded as follows:

  1. My only suggestion to improve the paper is to be careful about how some of the findings are reported. Using 95% significance means that there is still a potential 5% likelihood that the findings can be explained by chance. Therefore, with so many interactions and variables included, it is likely that SOME of the reported 'significant' findings might be a result of chance rather than genuine correlation. We agree with this and have pointed out the possibility of chance findings given the number of tests. We do prefer confidence intervals though as these give an indication of likely values of the odds ratios.
  2. Avoid using phrases such as "...WERE significantly less likely to..." We have revised the text to avoid this.
  3. It would also be helpful to add a sentence to the discussion to highlight the potential role of chance in some of the findings We have now include this.

Round 2

Reviewer 2 Report

I appreciate very  much your understandig what I was thinking about and changing being introduced. Now, it is better and I am very pleased to propose that Your text should be published.